# Diagnostically Challenging Subtypes of Invasive Lobular Carcinomas: How to Avoid Potential Diagnostic Pitfalls

**DOI:** 10.3390/diagnostics12112658

**Published:** 2022-11-01

**Authors:** Nektarios Koufopoulos, Ioannis S. Pateras, Alina Roxana Gouloumis, Argyro Ioanna Ieronimaki, Andriani Zacharatou, Aris Spathis, Danai Leventakou, Panagiota Economopoulou, Amanda Psyrri, Nikolaos Arkadopoulos, Ioannis G. Panayiotides

**Affiliations:** 1Second Department of Pathology, Medical School, Attikon University Hospital, National and Kapodistrian University of Athens, Chaidari, 12462 Athens, Greece; 2Medical Oncology Unit, 2nd Department of Internal Medicine-Propaedeutic, Medical School, Attikon University Hospital, National and Kapodistrian University of Athens, Chaidari, 12462 Athens, Greece; 34th Department of Surgery, Medical School, Attikon University Hospital, National and Kapodistrian University of Athens, Chaidari, 12462 Athens, Greece

**Keywords:** invasive lobular carcinoma, extracellular mucin production, lobular carcinoma with papillary features, lobular carcinoma with tubular elements, mucinous carcinoma, tubular carcinoma, differential diagnosis, solid papillary carcinoma, encapsulated papillary carcinoma

## Abstract

Invasive lobular carcinoma is the most common special breast carcinoma subtype, with unique morphological (discohesive cells, single-cell files, targetoid pattern) and immunohistochemical (loss of E-cadherin and β-catenin staining) features. Moreover, ILC displays a poor response to neoadjuvant therapy, a different metastatic pattern compared to invasive breast carcinoma of no special type, as well as unique molecular characteristics. In addition to the classic variant of invasive lobular carcinoma, several other well-recognized variants exist, including classic, alveolar, tubulolobular, solid, pleomorphic, signet-ring, and mixed. Furthermore, three novel variants of invasive lobular carcinoma, i.e., with extracellular mucin production, papillary features, and tubular elements, have been described during the last decade. We herewith focus on the unique morphological and immunohistochemical characteristics of these novel varieties of invasive lobular carcinoma, as well as differential diagnostic considerations and potential diagnostic pitfalls, especially when dealing with biopsy specimens.

## 1. Introduction

Invasive lobular carcinoma (ILC) is the second most common subtype of breast carcinoma following invasive breast carcinoma of no special type (NST), accounting for 10–15% of breast carcinoma [1]. It has unique clinical, imaging, histological, immunohistochemical, and molecular features.

While usually presenting as a mass-forming lesion, it may remain clinically undetectable.

Mammographic detection may be difficult [2]; ultrasonography is more sensitive but may underestimate tumor size [3]. Magnetic resonance imaging (MRI) may prove more helpful [4,5]. This underestimation of tumor size often results in positive surgical margins, thus requiring additional surgical procedures [6]. Occasionally, ILC may initially manifest as a distant metastatic lesion [7] and thus displays a different metastatic pattern from invasive breast carcinoma of NST [1]. ILC metastasizes less commonly to the lung but more frequently to the peritoneum, gastrointestinal and gynecologic systems [8,9,10,11].

Histologically, ILC is characterized by distinct morphological features, i.e., small cells lacking cohesion, forming single-cell files or arranged in a concentric pattern around existing ducts and lobular units (targetoid pattern), minimal stromal response, and the presence of a variable number of signet-ring cells [6]. Apart from the classic ILC, first described in 1941 by Foote and Stewart [12], a number of histological variants have so far been described, including alveolar [13], tubulolobular [14], solid [15], trabecular [13], signet-ring [16], pleomorphic [17,18] and mixed [1]. ILC is characterized by deleterious mutations in *CDH1* gene mapping in 16q22.1 accompanied by allelic loss of the remaining allele [6] and corresponding to a lack of E-cadherin and aberrant β-catenin immunostaining [19].

ILC usually has a luminal A molecular phenotype and is often estrogen receptor (ER)- and progesterone receptor (PR)-positive and HER−2 neu-negative, therefore being a prime candidate for hormonal therapy. It also shows a poor response to neoadjuvant therapy compared to invasive breast carcinoma of NST.

ILC presents specific molecular characteristics compared to other subtypes of invasive breast carcinoma.

Apart from the aforementioned variants, three new ones have recently been described, namely, ILC with extracellular mucin production, ILC with papillary features, and ILC with tubular elements. We herewith review the literature concerning these rare variants that may prove diagnostically challenging, focusing on their special morphological and immunohistochemical features, molecular characteristics, differential diagnostic considerations, and the potential diagnostic pitfalls, especially when dealing with biopsy specimens. This study aims to increase awareness of these rare variants of ILC.

## 2. Invasive Lobular Carcinoma with Extracellular Mucin Production

ILC with extracellular mucin production was first described by Rosa et al. in 2009 [20]. A review of the relevant literature has yielded 12 papers describing 39 cases in toto [20,21,22,23,24,25,26,27,28,29,30,31,32]. In some cases, important clinicopathological data, such as tumor size, lymph node, hormone receptor, Her−2 status, therapy, and follow-up information, are not mentioned.

The age ranged from 31 to 87 years (median 63). Most cases had a palpable or a radiographically detected lesion. The tumor size ranged from 7 to 100 mm (median 40 mm). Multifocality was present in several cases, one being a coexistent invasive breast carcinoma of NST [30]. One patient presented with skin ulceration [20] and another one with skin retraction and thickening, suggestive of inflammatory breast carcinoma [32]. Nodal status was positive in 17 cases.

Histologically, ILC with extracellular mucin production consists of two components: a mucinous (with multiple, relatively circumscribed, nodular foci of extracellular mucin and/or patchy extracellular mucin production) with irregular borders and a non-mucinous. The morphological and immunohistochemical characteristics of ILC with extracellular mucin production are shown in Figure 1.

The mucinous component percentage ranges from 5% to 95% of the total tumor area. In both mucinous and non-mucinous components, the architectural pattern can be of the classical, alveolar, and/or solid variants. Tumor cells are of small or intermediate size, pleomorphic or with apocrine morphology. Moreover, in most cases, a variable number of signet-ring cells can be identified in both areas. In many instances, concurrent lobular intraepithelial neoplasia was identified next to the ILC.

ER and PR were positive in 39/39 and 27/38 (one case does not mention PR status) cases, respectively. Her−2 neu was overexpressed in five tumors. The follow-up time ranged from 2 to 171 months (median 50). The clinicopathological data of all reported cases of ILC with extracellular mucin production are summarized in Table 1.

Soong et al. performed molecular analysis interrogating the full coding sequences of 447 genes for mutations and copy number variations and selected introns across for gene rearrangements in 13 cases with adequate material in their series [32]. As expected, alterations of *CDH1* at the gene level were present in all but one patient. Notably, concurrent *CDH1* mutations and 16q loss were detected in 11 of 13 cases. In the remaining two cases, one harbored 16q loss without detectable *CDH1* mutation, and the other harbored *CDH1* mutation without loss of a 16q arm. Interestingly, these two cases exhibited signet-ring cell morphology. The most common single-nucleotide variations (SNV) involved *CDH1* (10 cases), followed by *PIK3CA*, *POLQ*, *TP53*, *ERBB3* (3 cases each), *ERBB2*, and *RUNX1* (two cases each). Other genes with isolated SNV potentially clinically significant included *AKT1*, *FANCD2*, *GATA3*, *MAP3K1*, *MUTYH*, *PTEN*, *RB1*, and *SF3B1* [32].

Furthermore, five cases showed 17p loss, four 18q loss, and five 22q loss. *GATA3*, *FOXA1*, *CCND1*, *VEGFA*, *KAT6A*, and *POLB* recurrent gene amplifications and *ERBB2*, *CDK12*, *RUNX1*, *AURKA*, *ZNF217*, *ESR1*, *FGFR1*, *WHSC1L1*, isolated gene-level amplifications were identified. The above genomic analysis argues for the lobular type of this rare breast cancer variant.

## 3. Invasive Lobular Carcinoma (ILC) with Papillary Features

In 2016, Rakha et al. described three cases of ILC mimicking papillary carcinoma [33]; this was followed by four more single case reports [34,35,36,37]. This variant of ILC has been described under different terms such as “ILC mimicking papillary carcinoma” [33], “lobular breast cancer with solid growth pattern mimicking a solid-papillary carcinoma” [34], “ILC with papillary features” [35], “ILC with solid and encapsulated papillary carcinoma growth pattern” [36], “ILC mimicking encapsulated papillary carcinoma” [37].

Important clinicopathological data, including tumor size, lymph node, hormone receptor, Her−2 status, therapy, and follow-up information, are lacking in some of the aforementioned papers. In cases with available information, the age ranged from 61 to 86 years (median 73). All patients presented with a palpable lesion. Imaging findings (mammography and ultrasound) showed a well-circumscribed nodular mass in three cases [35,36,37] and a hypoechoic, highly vascularized mass in one case [34]. The tumor size ranged from 1.2 to 5.5 cm (median 2.8). There was no metastatic lymph node involvement in cases with available information.

Histologically, ILC with papillary features has, in most cases, two distinct components; the first consists of a nodular at least partially encapsulated tumor with a solid papillary growth pattern, fibrovascular cores, and focal hemosiderin deposition. In all but one case, a second component with the morphology of a classic variant ILC was found in close relation with the nodular neoplasm. The single case lacking a classic ILC component showed LCIS around the main tumor [36].

Immunohistochemically, all cases lack E-cadherin expression and are associated with loss of β-catenin [33,34] or aberrant (cytoplasmic) p120 [36,37] expression. GATA3 positivity supported a primary breast tumor [35,37]. Immunostains for various myoepithelial cell markers, such as p63, CK5/6, CK14, calponin, smooth muscle actin, and smooth muscle myosin heavy chain, were negative either within or in the periphery of the nodules. Immunostains for neuroendocrine differentiation markers (Chromogranin A, Synaptophysin, and CD56) were also negative where performed [33,35,36,37]. ER and PR showed intense staining, while HER−2 neu was uniformly negative. The morphological and immunohistochemical characteristics of ILC with papillary features are presented in Figure 2.

The three cases reported by Rakha et al. did not show local recurrence or metastasis within a median follow-up of 13 months [33]. Two additional cases mentioned that patients were alive with no evidence of disease after a follow-up of 8 and 10 months, respectively [36,37]. A summary of clinicopathological characteristics is seen in Table 2.

Christgen et al. examined the molecular profile of a case of ILC encompassing solid-papillary and classic variant ILC patterns [34], classified as luminal type B and luminal type A, respectively. Both variants shared common copy number profiles, including the loss of 6q, 11q, 13q, and 16q harboring *CDH1* and the gain of chromosome 1q, suggesting a common origin. Interestingly, sequencing analysis revealed a unique *CDH1* mutation resulting in a truncated E-cadherin protein in both components, further arguing for a clonal origin. Li et al. performed next-generation sequencing that revealed a frameshift mutation in exon 7 of *CDH1* [36].

## 4. Invasive Lobular Carcinoma (ILC) with Tubular Elements

This ILC variant displays tubular elements with P-cadherin expression. So far, there is only one manuscript (2020) describing 13 cases [38].

Clinically, the age ranged from 42 to 79 years (median 59). Clinical information concerning the imaging characteristics and tumor size was not mentioned. Five cases had lymph node involvement, two being micrometastases. Some cases were bifocal, multifocal, or bilateral. One case was ovarian metastasis.

Histologically, ILC with tubular elements is characterized by tumor cells arranged in several different architectural patterns seen in ILC (classic, trabecular, solid, and dispersed) admixed with tubular elements of variable shape and size (longitudinal, teardrop shaped, small round tubules with narrow lumina). Tubular elements represent a variable amount of total tumor area. Nuclear atypia is low or intermediate overall. In some cases, concurrent lobular carcinoma in situ can be identified. Figure 3 shows the histopathological and immunohistochemical features of a case of ILC with tubular elements recently diagnosed in our department.

Immunohistochemically, E-cadherin expression was strongly reduced in one and completely absent in 12 cases in both components, i.e., ILC and tubular. Despite this E-cadherin loss, most cases retained focal β-catenin expression. P-cadherin was expressed in 12 cases, with stronger staining in tubular elements and weaker or absent staining in the conventional ILC areas. All tumors were ER-positive, and Her−2 *neu*-negative; PR was positive in eleven cases. The tubular elements had a lower proliferative activity, as measured by Ki67 immunostaining. No information about follow-up time was available. Clinicopathological data of these cases are summarized in Table 3.

All tumors underwent molecular analysis. Eleven cases harbored *CHD1* mutations, while two lacked detectable *CDH1* mutations. Four patients exhibited bifocal, multifocal, and bilateral lesions with tubular elements present in a single focus. Molecular analysis in these cases showed almost always identical *CDH1* mutations in all foci, suggesting clonal relatedness of ILC with tubular elements and adjacent ILC with a conventional growth pattern.

Chromosomal analysis revealed gains on 1q, 8q, 11q13, and 16p and losses on chromosomes 8p, 11q, 16q, and 22q [38]. Notably, all cases examined exhibited either a copy number loss and loss of heterozygosity (LOH) or a copy number neutral LOH of 16q22.1 harboring *CDH1* gene [38]. In the same study, P-cadherin mRNA expression was assessed by quantitative real-time RT-PCR in two different regions, one with conventional morphology and the second consisting almost exclusively of tubules. The region with tubular morphology showed increased P-cadherin mRNA expression.

## 5. Discussion

All three aforementioned ILC variants may be diagnostically challenging, especially in limited biopsy material. This is due to the broad differential diagnosis of these variants, occasional sampling of the mucinous, solid papillary, or tubular component, and finally, lack of awareness. 

The differential diagnosis of ILC with extracellular mucin production includes invasive mucinous carcinoma of Capella type B, solid papillary carcinoma (SPC), mixed ILC and mucinous carcinoma, metaplastic matrix-producing carcinoma (MMPC), and polymorphous mammary adenocarcinoma.

Invasive mucinous carcinoma of the Capella B type lacks the classic morphological features of ILC and shows an expression of E-cadherin and β-catenin. SPC may show extracellular mucin production and can be associated with invasive mucinous carcinoma [39]; nevertheless, SPC lacks ILC morphology and shows an expression of E-cadherin and β-catenin. Also, Chromogranin and Synaptophysin are expressed in 50% of cases [40]. In mixed ILC and invasive mucinous carcinoma, immunostaining for E-cadherin will delineate the two different immunophenotypes. MMPC [41] and polymorphous mammary adenocarcinoma [42,43] also resemble ILC with extracellular mucin production since the former has a mucoid/matrix stroma, and the latter may simulate the single-file pattern of ILC [42,44]. Immunohistochemically, they may lack E-cadherin expression [42,45]. Moreover, both tumors lack ER expression, while MMPC also shows positive staining for S-100 with rare exceptions [41,46]. Finally, one should also not forget the possibility (albeit rare) of metastatic signet-ring cell carcinomas to the breast [25]. The morphological and immunohistochemical characteristics of ILC with extracellular mucin production and its mimickers are summarized in Table 4 and Table 5.

The differential diagnosis of ILC with papillary features includes solid papillary carcinoma (SPC), encapsulated papillary carcinoma (EPC), the solid or alveolar variant of ILC, an ILC-SPC collision tumor, confluent growth of LCIS, and colonization of a papillary neoplasm by lobular carcinoma in situ. Both SPC and EPC display cellular cohesion and are E-cadherin-positive. The solid and alveolar variants of ILC, the confluent growth of LCIS, and the colonization of a papillary neoplasm by LCIS lack the characteristic supporting fibrovascular cores. In the case of a collision tumor between an ILC and SPC, only the cells of the latter show cohesion. The morphological and immunohistochemical characteristics of ILC with papillary features and its mimickers are summarized in Table 6 and Table 7.

Tubular carcinoma, the tubulolobular variant of ILC, and mixed-type invasive breast carcinoma of NST and ILC may be considered in the differential diagnosis of ILC with tubular elements. Tubular carcinoma will not display a classic ILC pattern and shows strong E-cadherin, β-catenin, and membranous p120 expression. The mixed-type invasive breast carcinoma of NST and ILC will show positive staining for adhesion molecules in the tubules and an absence of staining in the ILC area [47]. The tubulolobular pattern of ILC shows strong E-cadherin and beta-catenin staining in both tubules and cells lacking cohesion [48,49,50]. Many authors consider this histological pattern a variant of invasive breast carcinoma of NST; however, the latest WHO edition still classifies this type as a variant of ILC [51]. 

The presence of tubular or pseudocribriform structures does not exclude a diagnosis of ILC [25], but an E-cadherin stain should be performed whenever these are present. 

The morphological and immunohistochemical characteristics of ILC with tubular elements and its mimickers are summarized in Table 8 and Table 9.

In the literature, cases of ILC with extracellular mucin production have occasionally been diagnosed as invasive mucinous carcinoma [25,28]. ILC with papillary features has been diagnosed as ductal carcinoma in situ, SPC, or encapsulated papillary carcinoma (EPC) in biopsy specimens [33,34]. In our own experience, we have misdiagnosed ILC with tubular structures as invasive breast carcinoma of NST in biopsy specimens (see Figure 3).

The result of misdiagnosis in these cases can be the misclassification of an ILC as a mucinous carcinoma with a favorable prognosis. In contrast, ILC clinical behavior is more aggressive than mucinous carcinoma, characterized as “initially indolent but slowly progressive” by Rakha et al. [52]. Misdiagnosis of ILC as SPC or EPC may have a worse impact on patient management since these entities are managed as in situ lesions, thus leading to suboptimal staging. The effect of diagnosing ILC with tubular structures as carcinoma of NST or mixed NST-ILC cannot be estimated, with no data concerning the former’s biologic behavior being available up to now. It is important for pathologists to keep in mind that extracellular mucin, solid papillary architecture, and tubular structures are not always associated with a ductal phenotype. Careful examination of lesions with the abovementioned characteristics on small biopsy specimens may reveal morphological findings suggestive of ILC (presence of signet-ring cells, lack of cellular cohesion, limited areas with a single-file pattern) that may otherwise be missed. Staining for E-cadherin, β-catenin, and p120 can be helpful in this context.

Concerning molecular findings, the analysis revealed in all three variants biallelic inactivation of *CDH1* due to concurrent *CDH1* mutation and complete or partial 16q arm loss [32,34,38], which is the genetic hallmark of ILC. Also, copy number alterations consistent with invasive carcinoma of NST, mucinous, SPC, or EPC were not found, thus confirming the morphologic and immunohistochemical findings.

Soong et al. concluded that ILC with extracellular mucin production cases with local recurrence and lymph node involvement bore more mutations with biologic significance compared to those lacking these features, this difference being not statistically significant, at least up to now [32].

In the case of ILC with papillary features presented by Christgen et al., molecular analysis confirmed a common clonal ancestry of the two morphologically distinct tumor components and that the papillary component was the result of subclonal evolution [34].

In what concerns ILC with tubular elements, molecular analysis showed, apart from the lobular phenotype, clonality of ILC with tubular elements with foci having a conventional ILC growth pattern in bifocal, multifocal, and bilateral tumors, and expression of P-cadherin in the mRNA and protein level [38].

## 6. Conclusions

In conclusion, we herewith present the clinicopathologic, morphological, immunohistochemical, and molecular characteristics of three rare variants of ILC, emphasizing the potential differential diagnostic issues. Additional cases need to be presented, and additional molecular studies should be performed to better understand both the biological potential and clinical behavior of these rare entities.

## Figures and Tables

**Figure 1 diagnostics-12-02658-f001:**
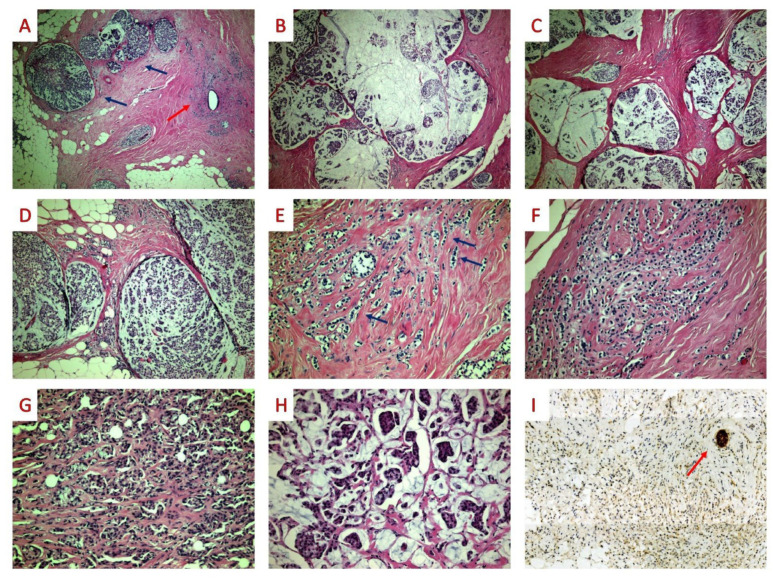
ILC with extracellular mucin production. ((**A**) 40×): The tumor consists of a mucinous (blue arrows) and a non-mucinous (red arrow) component. ((**B**,**C**) 40×): The mucinous component consists of multiple, relatively circumscribed, nodular foci of extracellular mucin. ((**D**) 100×): On higher power, tumor cells are arranged as single cells, clusters, pseudocribriform structures, or solid nests. ((**E**,**F**) 200×): The non-mucinous component shows typical ILC features such as a single-file pattern (blue arrows) and single cells lacking cohesion. ((**G)** 200×): In both areas without (1(**G)**) and with extracellular mucin production (1(**H**)), intracellular mucin and signet-ring cells can be seen. ((**I**) E-cadherin 200×) On immunohistochemistry, E-cadherin stain was negative; the red arrow points to positive internal control.

**Figure 2 diagnostics-12-02658-f002:**
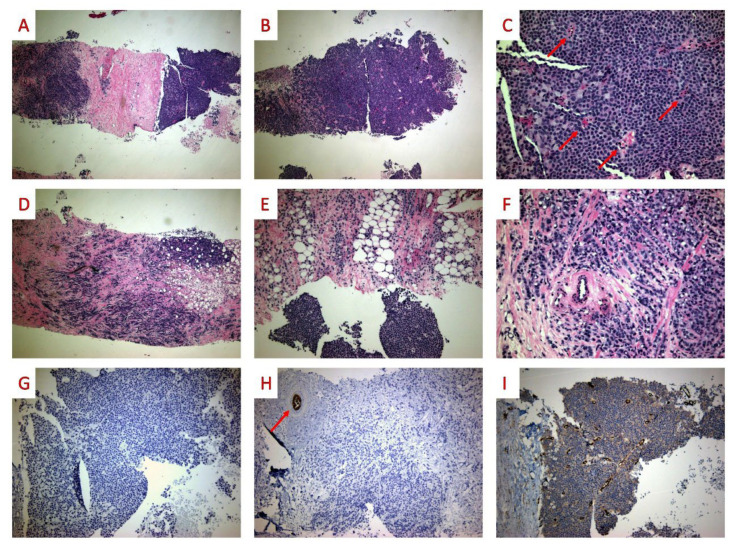
ILC with papillary features. ((**A**,**B**) 40×): The tumor consists of two components. The first had a solid architectural pattern. ((**C**) 200×): On high power examination, fibrovascular cores (red arrows) were observed. ((**D**,**E**) 40×): The second component shows a typical classic variant ILC morphology, with a single-file pattern and single-cell infiltration. ((**F**) 200×) On high-power examination, several signet-ring cells were visible. ((**G**,**H**) E-cadherin 100×): Immunohistochemically, E-cadherin was negative in both tumor components. Normal ducts and lobules served as an internal control (red arrow). ((**I**) β-catenin 100×): β-catenin was negative as well.

**Figure 3 diagnostics-12-02658-f003:**
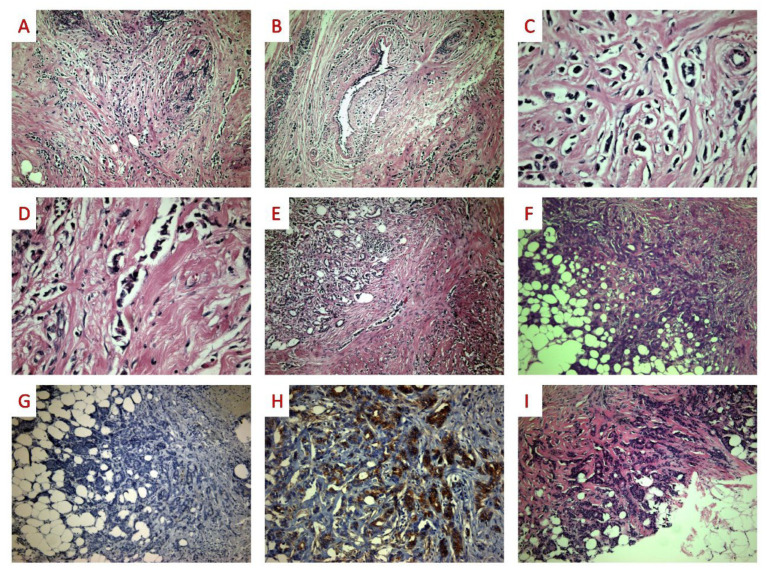
ILC with tubular elements ((**A**,**B**) 100×): Areas of conventional growth pattern consisting of single-cell files. ((**C**,**D**) 200×): On higher power examination, tumor cells are characterized by a lack of cohesion. ((**E**,**F**) 40×): Adjacent to the single-cell files, several tubular elements can be seen. ((**G**) E-cadherin 40): On immunohistochemistry, the tubular elements lack E-cadherin staining ((**H**) β-catenin 40×), but retain β-catenin expression ((**I**) 40×): This ILC variant may be misdiagnosed as well as differentiated NST or as tubular carcinoma, especially in a biopsy specimen consisting only of tubular structures.

**Table 1 diagnostics-12-02658-t001:** Clinicopathological findings of invasive lobular carcinoma with extracellular mucin production.

Case	Author	Year	Age	Size (mm)	Positive LΝ	pTNM	Nottingham Grade	ER/PR/HER2	Surgery	Adjuvant Therapy	Follow-Up (Months)
1	Rosa et al. [20]	2009	60	90	NA	pT3X	NA.	NA/NA/NA	M	NA	NA
2	Yu et al. [21]	2010	65	ΝA	1	pXN1	NA	P/N/P	BCS + SLNB	NA	NA
3	Haltas et al. [22]	2012	43	ΝA	1	pXN1	NA	P/P/N	M + ALND	NA	NA
4	Bari et al. [23]	2015	38	35	2n	pT2N1	NA	P/P/N	M + ALND	NA	NA
5	Gomez Macias et al. [24]	2016	60	9	0	pT1bN0sn	Grade 1	P/P/N	BCS + SLNB	RT + HT	NA
6	Cserni et al. [25]	2017	69	>24	1	pT2N1sn	Grade 2	P/P/N	M + SLNB	RT + HT	ANED 26
7	Cserni et al. [25]	2017	65	90	11	pT3N3	Grade 2	P/P/N	BCS > M	RT + HT + CT	DOD 40
8	Cserni et al. [25]	2017	71	46	0	pT2N0sn	Grade 2	P/P/P	BCS + SLNB	RT + HT + CT	ANED 29
9	Cserni et al. [25]	2017	62	80	10	pT3N3	Grade 2	P/P/Namp	BCS > M + ALND	RT	AWD 68
10	Cserni et al. [25]	2017	45	29	0	pT2N0sn	Grade 3	P/P/N	BCS + SLNB	RT + CT	ANED 2
11	Cserni et al. [25]	2017	56	22	0	pT2N0sn	Grade 2	P/P/N	M + SLNB	RT	ANED 11
12	Cserni et al. [25]	2017	75	30	7	pT2N2	Grade 2/3	P/N/N	EB > M + ALND	Neoadj CT	DOD 21
13	Cserni et al. [25]	2017	60	50	3	pT2N1	Grade 2	P/P/Namp	M + ALND	RT + HT + CT	NA
14	Boukhechba et al. [26]	2018	75	15	NA	NA.	NA	P/N/N	NA	NA	NA
15	Singh et al. [27]	2019	87	100	1	pT3N1	Grade 2	P/NA/N	NA	NA	NA
16	Singh et al. [27]	2019	72	16	NA.	NA	Grade 3	P/NA/N	NA	NA	NA
17	Singh et al. [27]	2019	70	>20	0	pT2N0	Grade 3	P/NA/N	NA	NA	NA
18	Singh et al. [27]	2019	77	8	0	pT1bN0	Grade 1	P/NA/N	NA	NA	NA
19	Baig et al. [28]	2019	67	60	0	pT3N0	Grade 3	P/N/Namp	BCS	NA	NA
20	Koufopoulos et al. [29]	2019	65	13 and 6	0	pT1cmN0sn	Grade 2	P/N/N	M + SLNB	RT + HT	ANED 10
21	Burky et al. [30]	2020	70	23	0	pT2mN0sn	Grade 3	P/P/P	M + SLNB	HT + CT	ANED 8
22	Hort et al. [31]	2022	62	70	1	pT3N1mi	Grade 2	P/P/Namp	M + ALND	HT + CT	NA
23	Soong et al. [32]	2022	71	15	0	pT1cmN0	Grade 2	P/N/N	NA	NA	DOD 114
24	Soong et al. [32]	2022	71	15	NA	pT1cmX	Grade 2	P/P/N	NA	NA	DOD 95
25	Soong et al. [32]	2022	31	30	3	pT2N1	Grade 3	P/P/P	NA	NA	ANED 35
26	Soong et al. [32]	2022	50	23	1	pT2N1	Grade 2	P/P/P	NA	NA	ANED 88
27	Soong et al. [32]	2022	69	7	0	pT1bmN0	Grade 2	P/P/N	NA	NA	NA
28	Soong et al. [32]	2022	68	70	3	pT3mN1	Grade 3	P/N/N	NA	NA	NA
29	Soong et al. [32]	2022	72	14	NA	pT1cmX	Grade 2	P/P/N	NA	NA	AWD 81
30	Soong et al. [32]	2022	70	73	1	pT3mN1	Grade 2	P/N/N	NA	NA	DOD 25
31	Soong et al. [32]	2022	77	52	0	pT3N0	Grade 2	P/P/N	NA	NA	ANED 3
32	Soong et al. [32]	2022	66	100	23	pT3N3	Grade 2	P/P/N	NA	NA	AWD 28
33	Soong et al. [32]	2022	59	28	1	pT2mN1mi	Grade 3	P/P/N	NA	NA	ANED 171
34	Soong et al. [32]	2022	45	18	0	pT1cmN0	Grade 2	P/P/N	NA	NA	ANED 167
35	Soong et al. [32]	2022	55	80	7	pT1cmN2	Grade 2	P/N/N	NA	NA	DOD 86
36	Soong et al. [32]	2022	66	16	0	pt1cN0	Grade 2	P/N/N	NA	NA	NA
37	Soong et al. [32]	2022	56	NA	NA	NA	Grade 2	P/P/N	NA	NA	ANED 42
38	Soong et al. [32]	2022	73	NA	NA	NA	Grade 2	P/P/N	NA	NA	NA
39	Soong et al. [32]	2022	50	NA	NA	NA	Grade 3	P/P/N	NA	NA	NA

Abbreviations: LN: lymph node; ER: estrogen receptors; PR: progesterone receptors; NA: not available; BCS: breast-conserving surgery; M: mastectomy; SLNB: sentinel lymph node biopsy; ALND: axillary lymph node dissection; RT: radiotherapy; HT: hormonal therapy; CT: chemotherapy: Neoadj: neoadjuvant therapy; ANED: alive no evidence of disease; AWD: alive with disease; DOD: died of disease.

**Table 2 diagnostics-12-02658-t002:** Clinicopathological findings of invasive lobular carcinoma with papillary features.

Case	Author	Year	Age	Size (mm)	Positive LΝ	pTNM	Nottingham Grade	ER/PR/HER2	Surgery	Adjuvant therapy	Follow-Up (months)
1	Rakha et al. [33]	2016	NA	28	0	pT2N0(sn)	NA	P/NA/N	R + SLNB	NA	ANED
2	Rakha et al. [33]	2016	NA	22	0	pT2N0(sn)	Grade 2	P/NA/N	M + SLNB	NA	ANED
3	Rakha et al. [33]	2016	NA	23	0	pT2N0(sn)	Grade 2	NA/NA/NA	R + SLNB	NA	ANED
4	Christgen et al. [34]	2017	74	55	NA.	pT3N0(sn)	Grade 2	P/P/N	M + SLNB	NA	NA
5	Motanagh et al. [35]	2020	86	30	No *	pT2X	NA	P/P/N	BCS	No *	NA
6	Li et al. [36]	2021	61	25	0	pT2N0(sn)	NA	P/P/N	M + SLNB	CT	ANED 10
7	Zheng et al. [37]	2022	73	12	NA.	pT1cX	NA	P/P/N	BCS	HT	ANED 8

Abbreviations: LN: lymph node; ER: estrogen receptors; PR: progesterone receptors; NA: not available; BCS: breast-conserving surgery; R: resection: M: mastectomy; SLNB: sentinel lymph node biopsy; HT: hormonal therapy; CT: chemotherapy; ANED: alive no evidence of disease; *: Patient refused further treatment;.

**Table 3 diagnostics-12-02658-t003:** Clinicopathological findings of invasive lobular carcinoma with tubular elements.

Case	Author	Year	Age	Size (mm)	Positive LΝ	pTNM	Nottingham Grade	ER/PR/HER2 Status	Surgery	Adjuvant Therapy	Follow-up (Months)
1	Christgen et al. [38]	2021	69	NA	NA	pT2mN0	Grade 3	P/N/Namp	NA	NA	NA
2	Christgen et al. [38]	2021	74	NA	NA	pT1cNx	Grade 2	P/P/N	NA	NA	NA
3	Christgen et al. [38]	2021	64	NA	NA	pM1 *	Grade 2	P/P/N	NA	NA	NA
4	Christgen et al. [38]	2021	50	NA	NA	pTxmNx	Grade 2	P/P/N	NA	NA	NA
5	Christgen et al. [38]	2021	61	NA	NA	pT1cmN0	Grade 2	P/P/N	NA	NA	NA
6	Christgen et al. [38]	2021	56	NA	NA	pT1bN0	Grade 2	P/N/N	NA	NA	NA
7	Christgen et al. [38]	2021	58	NA	NA	pT2N0	Grade 2	P/P/N	NA	NA	NA
8	Christgen et al. [38]	2021	50	NA	NA	pT2N1mi	Grade 2	P/P/N	NA	NA	NA
9	Christgen et al. [38]	2021	42	NA	NA	pT2mN1	Grade 2	P/P/N	NA	NA	NA
10	Christgen et al. [38]	2021	60	NA	NA	pT2N0	Grade 2	P/P/N	NA	NA	NA
11	Christgen et al. [38]	2021	59	NA	NA	pT1cN1mi	Grade 2	P/P/N	NA	NA	NA
12	Christgen et al. [38]	2021	75	NA	NA	pT3N1	Grade 2	P/P/N	NA	NA	NA
13	Christgen et al. [38]	2021	48	NA	NA	pT1cN1	Grade 3	P/P/N	NA	NA	NA

Abbreviations: LN: lymph node; ER: estrogen receptors; PR: progesterone receptors; NA: not available; Namp: 2+ immunohistochemical staining no amplification on FISH; * metastasis to the ovary.

**Table 4 diagnostics-12-02658-t004:** Morphological characteristics of invasive lobular carcinoma with extracellular mucin production and its differential diagnoses.

	ILCEMP	IMC	SPC	MMPC	PAdCa
Extracellular mucin	Present	Present	Present *	Absent	Absent
Single-file pattern	Present	Absent	Absent	Absent	Present
Targetoid pattern	Present	Absent	Absent	Absent	Absent
Signet-ring cells	Present	Present *	Present *	Absent	Absent
Mucoid matrix	Absent	Absent	Absent	Present	Absent
Fibrovascular cores	Absent	Absent	Present	Absent	Absent

Abbreviations: ILCEMP: invasive lobular carcinoma with extracellular mucin production; IMC: invasive mucinous carcinoma; SPC: solid papillary carcinoma; MMPC: metaplastic matrix producing carcinoma; PAdCa: polymorphous adenocarcinoma; *: occasionally present.

**Table 5 diagnostics-12-02658-t005:** Immunohistochemical characteristics of invasive lobular carcinoma with extracellular mucin production and its differential diagnoses.

	ILCEMP	IMC	SPC	MMPC	PAdCa
E-cadherin	Negative	Positive	Positive	Positive	Positive
ER	Positive	Positive	Positive	Negative	Negative
PR	Positive	Positive	Positive	Negative	Negative
S-100	Negative	Negative	Negative	Positive	NA
Chromogranin	Negative	Positive	Positive	Negative	Negative
Synaptophysin	Negative	Positive	Positive	Negative	NA

Abbreviations: ILCEMP: invasive lobular carcinoma with extracellular mucin production; IMC: invasive mucinous carcinoma; SPC: solid papillary carcinoma; MMPC: metaplastic matrix producing carcinoma; PAdCa: polymorphous adenocarcinoma; NA: not available.

**Table 6 diagnostics-12-02658-t006:** A summary of the most important distinguishing features of ILC with papillary pattern and its histological mimics.

	ILC with Papillary Features	SPC	EPC	Solid/Alveolar ILC	Confluent LCIS Growth	LCIS Colonization of a Papilloma
ILC component	Present	Absent	Present	Present	Absent	Absent
Fibrovascular cores	Present	Present	Present	Absent	Absent	Present
Fibrous capsule	Present	Absent	Present	Absent	Absent	Absent
Cellular cohesion	Absent	Present	Present	Absent	Absent	Absent

Abbreviations: ILC: invasive lobular carcinoma; SPC: solid papillary carcinoma; EPC: encapsulated papillary carcinoma; LCIS: lobular carcinoma in situ.

**Table 7 diagnostics-12-02658-t007:** Immunohistochemical characteristics of invasive lobular carcinoma with papillary features and its differential diagnoses.

	ILC with Papillary Features	SPC	EPC	Solid ILC	Confluent LCIS	LCIS Colonization of a Papilloma
E-cadherin	Negative	Positive	Positive	Negative	Negative	Negative
β-catenin	Negative	Positive	Positive	Negative	Negative	Negative
Chromogranin	Occasionally positive	Positive	Positive	Negative	Negative	Negative
Synaptophysin	Occasionally positive	Positive	Positive	Negative	Negative	Negative
p63	Negative	Negative	Negative	Negative	Positive *	Positive *

Abbreviations: ILC: invasive lobular carcinoma; SPC: solid papillary carcinoma; EPC: encapsulated papillary carcinoma; *: In the myoepithelial layer.

**Table 8 diagnostics-12-02658-t008:** Morphological characteristics of invasive lobular carcinoma with tubular elements and its differential diagnoses.

	Tubular Carcinoma	Tubulolobular Carcinoma	Mixed NST–Lobular Carcinoma	ILC with Tubular Elements
Tubules/glands	Small, round to ovoid or angular glands and tubules with open lumina	Small, round to angulated tubules	Variable morphology when present	Longitudinal, tear-drop shaped, small, round with narrow lumina
Single-cell files	Absent	Present	Present in ILC component	Present
Targetoid pattern	Absent	Present	Present in ILC component	Present

Abbreviations: ILC: invasive lobular carcinoma; NST: no special type.

**Table 9 diagnostics-12-02658-t009:** Immunohistochemical characteristics of invasive lobular carcinoma with tubular elements and its differential diagnoses.

	Tubular Carcinoma	Tubulolobular Carcinoma	Mixed NST–Lobular Carcinoma	ILC with Tubular Elements
E-cadherin	Positive	Positive	Positive in NST.Negative in ILC.	Negative
Beta-catenin	Positive	Positive	Positive in NST.Negative in ILC.	Negative
P-cadherin	Negative	Negative	Negative	Negative or weak in ILC. Positive in tubules

Abbreviations: ILC: invasive lobular carcinoma; NST: no special type.

## Data Availability

This article is a review and not an original study. All the references are listed.

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
