# Peer review of "Diagnostically Challenging Subtypes of Invasive Lobular Carcinomas: How to Avoid Potential Diagnostic Pitfalls"

_diagnostics, 2022, doi:10.3390/diagnostics12112658_

Round 1

Reviewer 1 Report

1) In the new era of interdisciplinary diagnosis and treatment of breast cancers it should be obligatory to correlate the histopathologic features of any subtype of breast cancer with the corresponding breast imaging findings (imaging biomarkers). The radiologists are involved in the diagnosis of each case and need to be able to correlate the low power, preferably large format histopathological structural changes with the corresponding imaging findings, including the mammographic / ultrasound / MRI images and the specimen X-ray.

2) It is unfortunate that the authors continue with the anatomically incorrect ductal-lobular dichotomy terminology that plagued four generations of pathologists. Using the same term, invasive "lobular carcinoma", to describe a spectrum of highly different breast malignancies implies that all of them have their origin in the acinar epithelium. 

In addition, the authors use the normal luminal cells of the acini as an "internal control" on Fig. 1, I and Fig. 2H, not realizing that their red arrows  prove that their so-called invasive lobular carcinoma has to have arisen from a source other than the epithelial cells within the "acini of the lobule". What justifies using the term "lobular" for these subtypes? 

3) Had the authors presented imaging-low power histopathologic correlation, they would have realized that most of their "ILC subtypes" such as the tubulolobular, solid, trabecular, etc. present on the mammogram as a tumor mass (stellate/spiculated or circular/oval). The clinical and imaging features of these subtypes are vastly different from the diffusely infiltrating, "classic variant of lobular carcinoma" (extensive architectural distortion on the mammogram with no tumor mass and thickening at clinical breast examination), still termed "lobular". Do the authors mean that the classic variant has its origin in the terminal ductal lobular units (TDLU) despite all these discrepancies? The breast cancers should be termed according to their site of origin, therefore those malignancies that originate from the acinar epithelium of the TDLU could be called either "lobular" or "acinar". But, the subtypes with the characteristic single files of the cancer cells, excessive fibrous tissue proliferation and very different imaging biomarkers must have a different site of origin (as the authors' Fig. 1 I and Fig. 2H show) and should not be termed "lobular". 

4) It is noteworthy that the immunohistochemical biomarkers in the "classic variant of invasive lobular carcinoma" are misleading since they imply a favorable outcome while these patients have a high risk of breast cancer death. This important aspect needs to be described for the members of the interdisciplinary breast cancer team.

Author Response

1) In the new era of interdisciplinary diagnosis and treatment of breast cancers it should be obligatory to correlate the histopathologic features of any subtype of breast cancer with the corresponding breast imaging findings (imaging biomarkers). The radiologists are involved in the diagnosis of each case and need to be able to correlate the low power, preferably large format histopathological structural changes with the corresponding imaging findings, including the mammographic / ultrasound / MRI images and the specimen X-ray.

We agree with reviewer nr.1 that radiologists are involved in the diagnosis of each case of breast carcinoma and that imaging findings have to be correlated to large-format histopathological structural changes.

2) It is unfortunate that the authors continue with the anatomically incorrect ductal-lobular dichotomy terminology that plagued four generations of pathologists. Using the same term, invasive "lobular carcinoma", to describe a spectrum of highly different breast malignancies implies that all of them have their origin in the acinar epithelium. 

In addition, the authors use the normal luminal cells of the acini as an "internal control" in Fig. 1, I and Fig. 2H, not realizing that their red arrows prove that their so-called invasive lobular carcinoma has to have arisen from a source other than the epithelial cells within the "acini of the lobule". What justifies using the term "lobular" for these subtypes?

We would like to thank the reviewer for giving us the opportunity to clarify this issue. Our manuscript is a review article and summarizes the knowledge on three very rare subtypes of invasive lobular carcinoma. We do not intend to discuss the ductal-lobular dichotomy. The latest data suggests that ILC is a distinct morpho-molecular breast cancer entity with well-defined clinical, histopathological, immunohistochemical, and molecular characteristics.

The use of normal breast epithelial structures as an internal control for immunohistochemistry is used in everyday practice in pathology. In fact, the use of internal controls is recommended by ASCO /CAP guidelines. We believe that the use of the term lobular is justified by three facts a) The morphological and immunohistochemical findings supported by the molecular findings in the cases presented. b) The fact that all authors reporting these subtypes use the term lobular, and 3) the fact that invasive lobular carcinoma is recognized in the latest (5th) edition of WHO breast tumours.

3) Had the authors presented imaging-low power histopathologic correlation, they would have realized that most of their "ILC subtypes" such as the tubulolobular, solid, trabecular, etc. present on the mammogram as a tumor mass (stellate/spiculated or circular/oval). The clinical and imaging features of these subtypes are vastly different from the diffusely infiltrating, "classic variant of lobular carcinoma" (extensive architectural distortion on the mammogram with no tumor mass and thickening at clinical breast examination), still termed "lobular". Do the authors mean that the classic variant has its origin in the terminal ductal lobular units (TDLU) despite all these discrepancies? The breast cancers should be termed according to their site of origin, therefore those malignancies that originate from the acinar epithelium of the TDLU could be called either "lobular" or "acinar". But, the subtypes with the characteristic single files of the cancer cells, excessive fibrous tissue proliferation and very different imaging biomarkers must have a different site of origin (as the authors' Fig. 1 I and Fig. 2H show) and should not be termed "lobular". 

We agree with reviewer 1 that the mammographic findings of some ILC subtypes are different than those of the classic variant. We do not mean that the classic variant has its origin in the TDLU. We use the term ILC, whether classic or any of its subtypes, based on the internationally established pathological criteria (combination of morphology and immunohistochemistry) as defined in the latest W.H.O. Classification of Breast Tumors, 5th edition, 2019)

4) It is noteworthy that the immunohistochemical biomarkers in the "classic variant of invasive lobular carcinoma" are misleading since they imply a favorable outcome while these patients have a high risk of breast cancer death. This important aspect needs to be described for the members of the interdisciplinary breast cancer team.

We would like to thank the reviewer for this comment. It is true that immunohistochemical biomarkers can be misleading and should not be interpreted without the proper morphology. For example, there are several triple-negative breast carcinomas with excellent behavior, such as adenoid cystic carcinoma and mucinous cystadenocarcinoma. In contrast, not all ER, PR positive, HER-2 negative invasive breast carcinomas necessarily have a good prognosis. Acknowledging this fact, we have described that invasive lobular carcinoma is regarded as “initially indolent but slowly progressive”. Our aim in this manuscript is not to discuss the clinical behavior of ILC but to review three recently characterized lobular carcinoma subtypes.

Reviewer 2 Report

Thank you for inviting me to review the manuscript entitled "Diagnostically Challenging Subtypes of Invasive Lobular Carcinomas: How to Avoid Potential Diagnostic Pitfalls" by Koufopoulos et al.

The scientific idea is clear.

Minor comments:

Where are the photos from? The quality could have been better.

Figure1 B,C – Magnificafion in my opinion is x100 not x40

H: intra? Or extracellular mucin

Tab. 4 i 5

Single-file pattern and Targetoid pattern in ILCEMP – I can’t see it

SPC ”Signet-ring cell features may be prominent” – WHO Breast tumours 5th ed.

SPC chromorganin and synaptophsin  half of cases  doi: 10.1097/PAS.0000000000000702.

Tab 6 . EPC  ILC component positive????

LCIS colonization of a papilloma  Fibrovascular cores  absent?????

Tubular carcinoma will not display a classic ILC pattern and shows strong E-Cadherin, β-catenin, and membranous p120 expression

Author Response

1) Where are the photos from? The quality could have been better.

The photos were taken from cases diagnosed in our Department (2nd Department of Pathology, “Attikon” University Hospital, Medical School, NKUA)

2) Figure1 B,C – Magnification in my opinion is x100 not x40

We have checked again, and the magnification is x40.

3) H: intra? Or extracellular mucin

We would like to thank the reviewer for noting this issue.  There is also extracellular mucin in figure 1H. In figures 1G and 1H we show that areas without (figure 1G) and with extracellular mucin (figure 1H) have intracellular mucin production. We have modified the figure legend accordingly.

4) Tab. 4 i 5

Could the reviewer clarify this issue?

5) Single-file pattern and Targetoid pattern in ILCEMP – I can’t see it

We thank the reviewer for giving us the opportunity to clarify this issue. We have replaced figure 1E with another one showing a single-file pattern.

6) SPC ”Signet-ring cell features may be prominent” – WHO Breast tumours 5th ed.

We would like to thank the reviewer for this comment. We do agree with reviewer 2. Signet ring cell features may be present in SPC, and sometimes it may be prominent. The presence of signet ring cells in SPC is mentioned in table nr. 4.

7) SPC chromorganin and synaptophsin half of cases  doi: 10.1097/PAS.0000000000000702.

We would like to thank the reviewer for this comment. We have added to the differential diagnosis discussion that SPC is positive in half of SPC cases. Chromogranin and Synaptophysin positive staining is also mentioned in table nr. 5

8) Tab 6 . EPC ILC component positive????

We have replaced the word positive with present.

9) LCIS colonization of a papilloma. Fibrovascular cores  absent?????

We would like to thank the reviewer for this comment.  When LCIS colonizes a papilloma, fibrovascular cores will not disappear. We have corrected this accordingly.

10) Tubular carcinoma will not display a classic ILC pattern and shows strong E-Cadherin, β-catenin, and membranous p120 expression

We have added to the manuscript that p120 expression is membranous.

Round 2

Reviewer 1 Report

This reviewer provided strong arguments for a complete rewrite of the manuscript. The authors have failed to comprehend the scope of my suggestions and have, instead, made a few cosmetic changes. They agreed upon the need of breast imaging-histopathologic correlation without even an attempt to provide it despite the request! I thoroughly disagree with their "internal control" explanation, even if they refer to "internationally established pathological criteria" as defined by the WHO. This reviewer considers the internationally established criteria clearly incorrect when the term invasive lobular carcinoma is used in the the presence of normal epithelial cells within the acini of the lobule. The authors owe the reader an explanation what justifies the use of "invasive lobular carcinoma" when they cannot prove that the malignancy has indeed its origin in the TDLU.  Unfortunately, this reviewer cannot condone the continuation of anatomically incorrect terminology.